# Knowledge, Attitude, and Perception towards Autism Spectrum Disorders among Parents in Sakaka, Al-Jouf Region, Saudi Arabia: A Cross-Sectional Study

**DOI:** 10.3390/healthcare12161596

**Published:** 2024-08-10

**Authors:** Bashayer Farhan ALRuwaili, Bader Abdullah T. Alrashdi, Ayesha Mallick, Thamer Alshami Marghel Alruwaili, Muhannad Faleh Alanazi, Hanan Farhan S. Alruwaili, Wael Faleh Alanazi, Waad Mudhhi Alanazi, Abdullah Fehaid Mukhlef Altaymani

**Affiliations:** 1Department of Family and Community Medicine, College of Medicine, Jouf University, Sakaka 72388, Saudi Arabia; barashid@moh.gov.sa (B.A.T.A.); amhaseeb@ju.edu.sa (A.M.); 2Department of General Administration of Health Programs, Diabetes Control Program, Ministry of Health, Riyadh 12542, Saudi Arabia; 3Department of Pediatrics, College of Medicine, Jouf University, Sakaka 72388, Saudi Arabia; tsruwaili@ju.edu.sa; 4Department of Internal Medicine, Division of Radiology, College of Medicine, Jouf University, Sakaka 72388, Saudi Arabia; mfalanazi@ju.edu.sa; 5Department of Neonatal Intensive Care Unit, Maternity and Children Hospital, Arara 73241, Saudi Arabia; hafaalruwaili@moh.gov.sa; 6Department of Intensive Care Unit, Aljouf Health Cluster, Alquryyat Hospital, Alquryyat 77453, Saudi Arabia; waelfa@moh.gov.sa (W.F.A.); walanazi6@moh.gov.sa (W.M.A.); 7College of Medicine, Jouf University, Sakaka 72388, Saudi Arabia; afmt1419@gmail.com

**Keywords:** autism spectrum disorder, knowledge, attitude, perception, females, early interventions

## Abstract

Parents are an essential element of family intervention for all children, including those with autism spectrum disorders (ASDs). We can better understand and address parents’ knowledge gaps about ASD through in-depth research and inquiry into parents’ current level of understanding, attitude, and perception. We aimed to assess the knowledge, attitude, and perception of ASD and influencing factors towards ASD among a group of parents with and without a child diagnosed with ASD in Sakaka, Al-Jouf Region, Saudi Arabia. Using the cross-sectional study design, information from the parents was gathered using a pretested questionnaire that included validated scales for measuring knowledge, attitudes, and perceptions related to ASD. The required number of participants was selected using the convenience sampling method. We used Spearman’s correlation test to determine the strength and direction of correlation between each domain. As a last step, we analyzed the influencing factors using binomial logistic regression. Among the 400 participants, 41.2% had high knowledge, 69.1% had a positive attitude, and 60.3% had a high perception of ASD. We found that knowledge was significantly higher among the parents with autistic individuals in the family (*p* = 038). The high and positive attitude was significantly greater among females (*p* = 0.010) and parents with high income (*p* = 0.007), and the perception was significantly associated with females (*p* = 0.037) and highly educated participants (*p* = 0.046). Furthermore, we found a positive correlation between knowledge, attitude, and perception. Overall, only less than half of the participants had a high knowledge of ASD. Hence, we recommend awareness-raising programs for the parents in this region. Furthermore, a prospective study involving parents from all provinces of Saudi Arabia is recommended.

## 1. Introduction

Autism spectrum disorder, also known as ASD, is a neurodevelopmental condition that is characterized by distinctive patterns in social communication and is characterized by a chronic, diverse nature [1,2,3] related to restricted, repetitive, or stereotypical behaviors [4]. For children with neurodevelopmental problems less than five years old, it is the leading cause of disability [3,4]. Among these severe neurological developmental abnormalities, therapeutic options are few. Even though the number of cases with ASD has been rising over the last 20 years, the diagnosis of the disorder is still based on subjective criteria, such as clinical observation and appraisal of individuals’ behavioral and developmental traits, rather than objective biomarkers. Unfortunately, many children do not obtain prompt diagnosis and therapy; many more are too young to articulate their emotions, and clinical signs are varied and complicated, making them difficult for parents to notice. Autistic individuals are valuable members of society because they possess strengths and a world outlook different from the rest of the population. They have varying abilities in functions like data analysis, ability to focus, creativity, or even memory, and they have, of course, made significant impacts in aspects of technology, art, or scientific enterprises [5,6]. However, there are still numerous difficulties observed in many families of autistic persons, such as financial, emotional, and social costs, that prove the necessity of appropriate support and resources [3,7].

Since its inception, ASD has seen a dramatic shift in both public understanding and information dissemination. Parents and other health care providers should stay up to date on handling ASDs due to the increasing number of children diagnosed with these disorders. Identifying children with ASD at an early stage and connecting them with the right intervention can have a lasting impact [8,9]. The global incidence of autism is somewhat below 1%. However, there is a wide variation across different countries [10,11]. The estimated average prevalence of autism spectrum disorder in Asia, Europe, and North America is 1%. The prevalence of ASD has been steadily rising in recent years, according to epidemiological studies. A recent review stated a consistent increase in ASD prevalence estimates within each geographical area since 2014 [11]. The disorder affects boys at a rate four to five times higher than girls [1,12]. In 2016, the Centers for Disease Control and Prevention (CDC) determined that approximately 1.68% of children in the United States, namely, those who were eight years old, were diagnosed with ASDs. This translates to 1 in 59 children [13]. Across the region around the Persian Gulf, the prevalence of ASD has been found to range from 0.14 to 2.9%; meanwhile, across Asia, the prevalence has been recorded to be 3.9% [1].

There has been no conclusive research on the causes of ASD. Questions such as its origin, the role of genes and the environment, and the interplay between these factors are all under investigation, as are the many cognitive, behavioral, and developmental traits found in those who have it [4]. Immunological, perinatal, and metabolic variables have been proposed as potential causes of ASD, according to a large body of research [1]. The severity of ASD can range from mild to severe, depending on the specific form of the disorder. In cases of extreme severity, it is essential to offer substantial assistance and continuous attention. Rigidity and resistance to change are hallmarks of ASD in youngsters. Social interactions, motor functions, intellect, and verbal and nonverbal social communication abilities are all negatively impacted in autistic children. Atypical sensory reactions, repeated habits, and strange interests are other hallmarks of these patients.

Inadequate education regarding autism diagnoses is one of the critical challenges faced by the families of children with ASD. Thus, these parents seldom feel confident about their ability to identify ASD at the early stage that supports their child’s development. This issue was further complicated by the widespread misconceptions among the general population and among parents of children with autism [14,15]. Such misconceptions hinder the appropriate education and accessibility of resources for families, and, thus, the latter lose confidence in their ability to promote the proper development of the child. A study conducted in 2023 found a lack of awareness and knowledge regarding ASD among the general public of Jordan. Additionally, they demonstrated that females, young age, and participants with less income had higher levels of knowledge regarding ASD [16]. On evaluation knowledge, attitude, and perception, a study from Greece found that their participants had a very good knowledge of ASD [17]. These findings indicate that the general population’s awareness and understanding of ASD vary across the region, and there is a need for region-specific data. 

In addition to this, it is pivotal to assess the current knowledge gaps of parents, which is the base for applying the recent advancement in the diagnosis and intervention of autistic persons. For instance, current investigations evidence that the application of artificial intelligence, including but not limited to machine learning algorithms, may be an effective pre-screening technique for ASD [18,19]. Besides that, recent developments like vision-based assistance systems for observing stereotyped manners and image analysis for identifying spasmodic manners are other achievement [18,20]. Therefore, recognizing parents’ current thoughts and knowledge deficits related to ASD in the first move can assist us in introducing them to recent advancements in the field of autism.

Parents perceived the treatments for ASD as being too expensive and out of reach. Parents’ experiences and access to effective treatment were impacted by the shortage of qualified specialists in the area of ASD [21,22]. When it comes to family intervention, parents are definitely an essential component. Treatment, prognosis, and diagnosis of ASD are all impacted by parents’ level of understanding and awareness of the disorder [23]. We can better understand and address parents’ knowledge gaps about ASD through in-depth research and inquiry into parents’ current level of understanding, attitude, and perception. This could allow us to create more targeted parental training programs. Most importantly, it can also help us identify whether knowledge training is helpful [14]. Hence, it is necessary to determine the magnitude of the issue and the elements that contributed to it. Furthermore, there is a lack of region-specific data that provide valuable insights into the sociocultural and educational factors influencing parental understanding and acceptance of ASD. Therefore, this study was planned to assess the parents’ knowledge, attitude, perception, and associated factors of ASD in Sakaka, Al-Jouf Region, Saudi Arabia. Furthermore, the authors aimed to determine the strength and direction of each domain. 

To achieve these objectives, the authors formed the following measurable research questions:What is the current level of knowledge about ASD among parents in Sakaka, Al-Jouf Region?What are the attitudes and perceptions of parents towards individuals with ASD?What is the strength and direction of the correlations between parents’ knowledge, attitudes, and perceptions regarding ASD among the study participants?Which factors are significantly associated with knowledge, attitudes, and perceptions of ASD?

## 2. Materials and Methods

### 2.1. Study Design

This cross-sectional study was conducted from November 2023 to May 2024.

### 2.2. Study Setting

The present study was conducted in public places such as malls, festivals, and parks in Sakaka, Al-Jouf region, Saudi Arabia.

### 2.3. Inclusion and Exclusion Criteria

We included a group of parents (Saudi nationals) with and without a child diagnosed with ASD who were aged from 18 to 49 years of age from Sakaka, Al-Jouf region, Saudi Arabia. Al-jouf region is in the northern part of Saudi Arabia with a population of approximately half a million. The population in this region is characterized by a strong agricultural background and follows a strong traditional valued life. Any individual belonging to the medical profession, who received formal autism-related training, was certified with psychiatric disorders, and unwilling participants were excluded from the study.

### 2.4. Sample Strategies

The sample was calculated using the equation of *n* = z^2^p (1 − p)/e^2^, based on the degree of assessment of parents’ knowledge towards ASD as 50% (p). As a rule of thumb, this value was considered by the research team to obtain the maximum number of participants to obtain a valid conclusion. Other values applied in the formula were margin of error (e) 5%, 95% confidence interval (z = 1.96), and power of the study as 80%. After careful calculations using these values, we obtained 384 as the minimum required parents. Participants were invited by the research team using convenience sampling from public places such as parks, malls, and masjids by the research team. We recruited participants from diverse locations. This method was applied for practicality to ensure broader outreach and efficiency. Once potential participants were identified, they were approached by the research team members and provided with a brief overview of the study. This overview included the purpose of the research, the steps involved, the potential risks and benefits, and the importance of their participation. We obtained informed consent from the participants after the briefing.

### 2.5. Data Collection Methods and Tools

We obtained ethical approval to conduct this study from the Local Committee of Bioethics (LCBE) of Jouf University (Approval no: 2-07-45, Dated: 17 March 2024). The data were collected from the participants using a structured self-administered questionnaire (Arabic). This questionnaire was prepared based on open sources existing literature and focus group discussion with the family medicine, child health, and psychiatry department faculties (content validity) [15,24,25]. Other validities, including face validity and reliability, were tested during the pilot study that was carried out among 30 participants. All the parents who participated were informed that the structured data collection tool was simple and easy to understand. The average time for completing the survey was around 8 min. Finally, we performed exploratory factor analysis for construct validity, and the results supported the three-factor structure. The authors repeated the survey to the same pilot study participants after two weeks. The test–retest reliability coefficients were 0.88 for knowledge, 0.81 for attitudes, and 0.84 for perceptions. Cronbach’s alpha was calculated for each domain of the questionnaire (knowledge, attitudes, and perceptions) to assess internal consistency. Cronbach’s alpha values for the data collection tool were 0.77, 0.81, and 0.91 for the knowledge, attitude, and perception domains. The data collection form consisted of four sections, as mentioned below:Section 1.Socio-demographic characteristics include age, sex, the income of the family, the educational level, marital, and occupation status of parents. Furthermore, we asked for the details of the total number of children, the number of children less than 10 years of age, the youngest child’s age, and any child with ASD in their family.Section 2.Knowledge assessment about ASD: Parents’ knowledge of the symptoms and behavior of children with ASD, typical child development, and the treatability of ASD (10 items).Section 3.Attitudes were tested through statements about the parents’ attitudes and feelings towards ASD children (10 items).Section 4.Perception assessment: Parents were asked about their perception toward some characteristics of ASD children (10 items).

Each question in the knowledge section of the questionnaire could be answered with yes, no, or do not know the answer. Correct answers were given 1 point, and incorrect or do not know were given zero points. At the same time, the attitudes and perception section had five options (ranging from strongly agree to strongly disagree). The score for each question ranged from 1 to 5. The outcome was measured by computing scores in each section, including knowledge, attitudes, and parents’ perceptions toward ASD. Furthermore, we categorized the total points of each domain into low (<60% of overall points), medium (60 to 79% of overall points), and high (≥80% of overall points). This category was based on Bloom’s cutoff points that were used by numerous studies [26,27]. 

### 2.6. Data Analysis

The collected data were tabulated and statistically analyzed using Microsoft Excel and Statistical Package for the Social Sciences software version 24.0 (SPSS, V 24.0). Quantitative data were expressed in numbers and percentages. We performed the Spearman rank correlation test to determine the strength and direction of the association. This test was performed as our data did not meet the normality assumption that was identified through Shapiro–Wilk’s test. Shapiro–Wilk’s test revealed that all three domains (knowledge, attitude, and perception) scores were skewed distribution (*p* < 0.001). A multivariate analysis using binomial logistic regression analysis was performed to determine the associated factors. In logistic regression analysis, we utilized the enter method to include all study (exposure) variables in the model simultaneously. In this method, the outcome variable was set as low/medium (combined) and high. A *p*-value less than 0.05 was considered a significant variable associated with knowledge, attitude, and perceptions. 

## 3. Results

During the data collection period, we approached 449 eligible participants, and 398 (above the minimum required sample size) agreed to participate in the survey. Hence, the response rate was 88.6%. More than half of the participants (52%) were between 18 and 30 years old. More than half of parents (50.5%) were males. Participants who had more than 7000 SAR constituted 48.7%. About 61.6% were in university or graduate education. The majority of participants (89.7%) were married. Nearly half of the parents (51.5%) were government employees, while one-third of them (33%) were unemployed. The participants who had children less than 10 years of age were 73.1%. The number of children ≤ 3 years was 89.9%. Only 11.3% of parents had children suffering from ASD (Table 1).

The knowledge of parents about ASD is described in Table 2. More than half of parents (56.8%) had wrong answers about ASD as a result of poor parenting. Only 35.2% of parents answered correctly about symptoms of autism. A total of 56.3% answered correctly that autistic children will not maintain eye-to-eye contact. More than half of the parents (59.3%) knew that social skills training is the key component of early intervention for ASD; only 26.1% of the parents knew that most children with autism also display comorbidities with intellectual disabilities. Regarding cures for autistic individuals, more than three-fourths (76.1%) answered wrongly that ASD can be completely cured.

Table 3 shows the distribution of parents according to their attitudes about ASD. More than half of parents (52.1%) believed that individuals with autism can lead fulfilling lives. Moreover, 70.6% of parents agreed and strongly agreed about their thoughts regarding the importance for schools to implement inclusive practices for students with autism. More than two-thirds (68.9%) of parents agreed and strongly agreed about supporting public awareness campaigns to reduce the stigma associated with autism. About 68.3% of the parents believed that children with ASD need additional support in the workplace. More than two-thirds of parents (70.1%) had agreed and strongly agreed attitude about the importance of early intervention for ASD children.

Table 4 shows the distribution of parents by their perception of ASD. More than half of parents (57.3%) agreed and strongly agreed about enough awareness of the challenges faced by individuals with Autism. Only 4% of the parents strongly disagree that diagnosing a child with autism will lead to discrimination against the child. Nearly one-third of all strongly agree, agree, and neutral answers about that media represents positive about autism. Furthermore, about half of the participants strongly agreed with the statements, “Important of early intervention for ASD children (51.5%)”, “I think it is important for schools to implement inclusive practices for students with autism (48.5%), and “I support public awareness campaigns to reduce the stigma associated with autism (47.5%)”.

Table 5 presents the distribution of parents according to their knowledge, attitude, and perception of ASD. Parents who had high, medium, and low levels of knowledge constituted 41.2%, 24.4%, and 34.4%, respectively. As regards their attitude, high, medium, and low levels constituted 69.1%, 24.9%, and 6% respectively. Moreover, parents who had high, medium, and low levels of perception constituted 60.3%, 30.2%, and 9.5%, respectively.

The relationship between the level of knowledge, attitude, and perception score is shown in Table 6. A statistically significant positive correlation existed between parents’ knowledge, attitude, and perception scores. The difference was statistically significant (*p* = 0.001).

Table 7 shows the relationship between the socio-demographics of the parents and their knowledge about ASD. There was a statistically significant association between the children who suffer from ASD in the family and the degree level of knowledge of parents (adjusted odds ratio [AOR] =1.84; 95% confidence interval [CI] = 1.13–2.63; *p* = 0.038).

Table 8 depicts the relationship between the socio-demographics of the parents and their attitudes toward ASD. There was a statistically significant association between the gender (AOR = 1.76; 95% CI = 1.14–2.71; *p* = 0.010) and income (AOR = 1.46; 95% CI = 1.03–1.74; *p* = 0.007) with the attitude of parents about ASD. No other background characteristics of the study population were significantly associated with the knowledge domain.

Table 9 shows the relationship between the socio-demographics of the parents and their perceptions about ASD. There was a statistically significant association between gender (AOR = 1.54, 95% CI = 1.03–2.31; *p* = 0.037) and education status (AOR = 1.48; 95% CI = 1.07–2.25; *p* = 0.046) with the perception of parents about ASD.

## 4. Discussion

The current study assessed the parents’ knowledge, attitude, perception, and associated factors of ASD in Sakaka, Al-Jouf Region, Saudi Arabia. The present study found that less than half of the participants had a high knowledge of ASD. This finding explains the need for targeted educational interventions to improve awareness and understanding of ASD among parents. On the other hand, the studies conducted in Saudi Arabia in 2022 and 2023 showed a weak level of knowledge among parents about ASD [24,28]. Moreover, another study conducted in Karachi, Pakistan, among parents of ASD children found a poor level of knowledge [29]. The study found that the knowledge of the parents increases with an increasing number of children who are suffering from ASD in the family. This finding is consistent with the study conducted by Irene Gomes and others among Spanish parents [30]. Anther’s study revealed similar results to those of a study done in Malaysia (2022) [31]. Some authors from nearby Arab countries explored public awareness and attitudes towards autistic individuals and revealed interesting findings [16,32]. For instance, a recent study from Jordan by Abuhamdah SMA et al. reported that even though their study participants showed a moderately positive attitude toward autistic individuals, their awareness of ASD was poor [16]. These insights highlight the importance of targeted educational interventions to improve understanding and support for families with children diagnosed with ASD. Factors contributing to differences in findings across different countries can be attributed to geographical, methodological, and socio-demographic characteristics. The present study used a validated tool that was adapted from different studies.

The current research showed that there is a correlation between the amount of knowledge, attitude, and perspective among parents and a variety of parameters, including gender, income, educational status, and the presence of children in the family who are diagnosed with ASD. This confirms the previous study in Saudi Arabia’s Aseer Region (2023), which discovered a correlation between parents’ knowledge and their children’s gender and educational level [28]. It is worth mentioning here that the education system in Saudi Arabia is mandatory primary and secondary education, with increasing incorporation of special education programs the intellectually disabled children [33]. There was also no statistically significant correlation between parents’ knowledge and socio-demographic variables in this study, including age, gender, income, education, marital status, occupation, family size (including number of children under 10 years old), and number of children in the family. This could be attributed to the difference between the sociocultural and behavioral characteristics of the community. Therefore, understanding these sociocultural influences in the Aljouf region is crucial for designing effective educational and support programs that are culturally appropriate and impactful. The study found that the knowledge was not influenced by the differences between the ages of parents. This finding is consistent with a study conducted in Pakistan, which showed no statistically significant association between age, marital status, and parents’ knowledge [29]. Moreover, many of the studies conducted among different developing countries found the same findings, such as a study performed in Saudia Arabia (2021) [34], Saudi Arabia (2023) [28], and Malaysia (2022) [31]. On the other hand, a survey conducted among Chinese families in 2019 showed that the age of the parents, income, and education status had statistically significant associations with the knowledge of parents [14]. This highlights the need for culturally tailored approaches to ASD education and awareness programs. Further, a descriptive study conducted in Egypt (2023) supported these findings, which found a statistically significant association between the age of the parents, education status, and occupation with the knowledge of parents [35]. Moreover, many studies consistent with this study, such as a study performed in Saudi Arabia (2021) [34] and Malaysia (2022) [31]. Another study conducted in Karachi, Pakistan, among parents of ASD children found a statistically significant association between gender, income, education status, and occupation with the knowledge of parents [29]. Insights from studies in diverse settings underscore the importance of understanding local sociocultural contexts when designing interventions to enhance parental knowledge and support for children with ASD.

Regarding the attitude of the parents toward ASD. There was a statistically significant association between ASD and the attitude of the parents. The difference was statistically significant (*p* = 0.001). The findings of this study indicated that the overall attitude rate among parents was 69.1%. This is consistent with the study conducted in Saudia Arabia (2021) [34]. Moreover, another study conducted in Spain (2023) supported this finding [36]. These findings were not in line with other studies done in India [37] and Saudi Arabia (2021) [34]. The observed improvement in the attitude of the parents toward ASD is a result of the widespread use of social media, the availability of information, and the ease of obtaining it on the internet. These factors, therefore, account for the observed higher level of parents’ attitudes towards ASD in Sakaka, Al-Jouf Region. There is a possibility of circulating good messages and support; good sources of information can help equip the parents with knowledge; and there is always easily accessible information that can help parents acquire whatever resource they need without much hassle. Altogether, these factors help create a better informed and empathetic community, which in turn affects the perception of ASD. However, the possibility of obtaining the wrong pieces of information through social media also cannot be ignored. Hence, healthcare workers and concerned authorities should take the necessary steps to guide the parents in obtaining proper information.

The findings of this study indicated that the overall perception rate among parents was 60.3%. The current finding is consistent with the studies conducted in a general hospital in Porto, Portugal (2021) [38], US (2019 [39], China (2022) [40]. However, a cross-sectional study that spanned from 2021 to 2022 in Saudi Arabia used an online survey tool to investigate how the general population there knew about typical child development and autism spectrum disorder (ASD). The results showed that people there knew very little about ASD, with a mean score of 5.9 (SD: 3.1), or 34.7% of the best possible score [24]. By getting in the way of appropriate ASD education and treatment, misconceptions about the disorder can have a detrimental impact on ASD-diagnosed families. Consequently, it is of the utmost significance to investigate the notions that parents have regarding the causes of ASD [25]. There was a statistically significant association between gender (*p* = 0.037) and education status (*p* = 0.046) with the perception of parents about ASD. The findings in this study are consistent with the survey conducted in the state of Espírito Santo, Brazil (2021) [41]. In the current study, more than half of parents (57.3%) agreed and strongly agreed about enough awareness of the challenges faced by individuals with Autism. Discrimination against a kid diagnosed with autism would increase, according to just 4% of parents who strongly disagree. Nearly one-third of all strongly agree, agree, and neutral answers about that media represents positive about autism. A systematic review carried out in 2022 among parents toward perceptions of ASD in Latinx and black sociocultural contexts supports these findings [42]. These insights emphasize the need for targeted educational campaigns and inclusive media representation to foster greater understanding and support for individuals with ASD. Several governmental and non-government organizations, such as the World Health Organization, have developed and circulated numerous offline and online training programs [43,44,45]. Few studies demonstrated an improvement in parental knowledge and perceptions after the effective implementation of parental and caregiver training programs. Interestingly, a study by Kenworthy L et al. in 2023 reported that online training was as effective as in-person training [44]. These parental training programs, especially culturally rooted programs, have been proven to improve the levels of parent’s empowerment [43,46,47].

## 5. Limitations of the Study

This is the first kind of study conducted in this region. Furthermore, we used a standard methodology. However, some limitations need to be considered while reading this research paper. First, the parent sample was obtained in a public place using convenience sampling, and several parents may not visit the mall or park frequently, which was neglected in this study. This sampling method may introduce sampling bias, as the sample may not fully represent the broader population of parents in Sakaka, Al-Jouf Region. The self-selection bias among the study participants could be another limitation of the study. Next, the cross-sectional study design used in this study limits our ability to draw conclusions about causal relationships or changes over time. Finally, the generalizability of this study’s findings is limited as wide sociocultural variation across Saudi Arabia exists.

## 6. Conclusions

Overall, less than half of the participants had a high knowledge of ASD. We found females had a high attitude and perception towards ASD. Having a higher education is also another factor associated with the perception of parents towards ASD. Our findings indicate there are notable gaps that need to be addressed through targeted educational interventions and support programs. Hence, we recommend awareness-raising programs for the parents of this region. This awareness-raising program can be conducted to help the target population improve their knowledge of ASD. By implementing these programs, policymakers can contribute to the development of comprehensive support systems for autistic individuals and their families. Furthermore, a prospective study involving parents from all provinces of Saudi Arabia is recommended to obtain more in-depth information to improve and strengthen ASD treatment programs.

## Figures and Tables

**Table 1 healthcare-12-01596-t001:** Background characteristics of the participants (*n* = 398).

Variable	Frequency	Proportion
Age group		
18 to 30	207	52.0
31 to 40	120	30.2
41 and above	71	17.8
Gender		
Male	201	50.5
Female	197	49.5
Income		
Less than 5000 SAR	105	26.4
5000 to 7000 SAR	99	24.9
More than 7000 SAR	194	48.7
Education status		
Up to high school	153	38.4
University/Graduate studies	245	61.6
Marital status		
Currently married	357	89.7
Divorced	41	10.3
Occupation		
Government	205	51.5
Private	60	15.1
Unemployed	133	33.4
Children less than 10 years		
No	107	26.9
Yes	291	73.1
Number of children		
≤3	358	89.9
>3	40	10.1
Does your family have a child diagnosed with ASD?		
No	353	88.7
Yes	45	11.3
Youngest child age (in years)		
0 to 6	356	89.4
More than 6	42	10.6

**Table 2 healthcare-12-01596-t002:** Participants’ responses in the knowledge category (*n* = 398).

Items	Correct Answer		Wrong Answer	
	*n*	%	*n*	%
Autism Spectrum Disorders are a result of poor parenting	172	43.2	226	56.8
Many children show symptoms of autism	140	35.2	258	64.8
Children with autism will not maintain eye-to-eye contact	174	43.7	224	56.3
Do children with autism usually grow up to be adults with schizophrenia	92	23.1	306	76.9
Social skills training is the key component of early intervention for autism spectrum disorder	236	59.3	162	40.7
Autism can be completely cured	95	23.9	303	76.1
Children with autism often have speech delays	231	58.0	167	42.0
Males get autism more than females	117	29.4	281	70.6
Important of early intervention for ASD children	239	60.1	159	39.9
Most children with autism also display comorbidities with intellectual disabilities	104	26.1	294	73.9

**Table 3 healthcare-12-01596-t003:** Participants responses in attitude section (*n* = 398).

Item	Strongly Agree *n* (%)	Agree *n* (%)	Neutral *n* (%)	Disagree *n* (%)	Strongly Disagree *n* (%)
I believe that individuals with autism can lead fulfilling lives	109 (27.4)	99 (24.9)	124 (31.2)	55 (13.8)	11 (2.8)
I feel comfortable interacting with autistic people	67 (16.8)	113 (28.4)	162 (40.7)	44 (11.1)	12 (3.0)
I think it is important for schools to implement inclusive practices for students with autism.	193 (48.5)	84 (21.1)	94 (23.6)	22 (5.5)	5 (1.3)
I believe that people in my community are well-informed about Autism	101 (25.4)	95 (23.9)	118 (29.6)	65 (16.3)	19 (4.8)
I support public awareness campaigns to reduce the stigma associated with autism	189 (47.5)	85 (21.4)	98 (24.6)	23 (5.8)	3 (0.8)
additional support in the workplace.	186 (46.7)	86 (21.6)	95 (23.9)	25 (6.3)	6 (1.5)
Important of early intervention for ASD children	205 (51.5)	74 (18.6)	93 (23.4)	23 (5.8)	3 (0.8)
I feel that my community is supportive of families with children who have autism	147 (36.9)	93 (23.4)	116 (29.1)	32 (8.0)	10 (2.5)
Autistic should have equal opportunities in education in regular school.	155 (38.9)	99 (24.9)	109 (27.4)	25 (6.3)	10 (2.5)
I am open to learning more about autism to better understand and support them	194 (48.7)	94 (23.6)	84 (21.1)	20 (5.0)	6 (1.5)

**Table 4 healthcare-12-01596-t004:** Participants’ responses in the perception section (*n* = 398).

Items	Strongly Agree *n* (%)	Agree *n* (%)	Neutral *n* (%)	Disagree *n* (%)	Strongly Disagree *n* (%)
I think there is enough awareness about the challenges faced by individuals with Autism	139 (34.9)	89 (22.4)	118 (29.6)	43 (10.8)	9 (2.3)
I think that diagnosing a child with autism will lead to discrimination against the child	73 (18.3)	105 (26.4)	147 (36.9)	57 (14.3)	16 (4.0)
I believe media represents positive about autism	121 (30.4)	122 (30.7)	126 (31.7)	23 (5.8)	6 (1.5)
Autism is unrecognized and often missed in general practice	79 (19.8)	103 (25.9)	144 (36.2)	54 (13.6)	18 (4.5)
I think healthcare professionals treat autistic people fairly	125 (31.4)	120 (30.2)	129 (32.4)	19 (4.8)	5 (1.3)
I am confident of the accuracy of autism screening methods	105 (26.4)	119 (29.9)	138 (34.7)	27 (6.8)	9 (2.3)
I perceive that autistic people are well-represented in educational materials	120 (30.2)	116 (29.1)	138 (34.7)	19 (4.8)	5 (1.3)
I perceive that autistic people are welcomed by my community	128 (32.2)	111 (27.9)	121 (30.4)	32 (8.5)	4 (1.0)
I perceive that there is sufficient social support for families with children who have Autism	141 (35.4)	96 (24.1)	128 (32.2)	27 (6.8)	6 (1.5)
I believe that the community is adequately informed about the unique strengths and abilities of individuals with Autism	112 (28.1)	112 (28.1)	120 (30.2)	45 (11.3)	9 (2.3)

**Table 5 healthcare-12-01596-t005:** Knowledge, attitude, and perception (KAP) categories (*n* = 398).

Domain	Low *n* (%)	Medium *n* (%)	High *n* (%)
Knowledge	137 (34.4%)	97 (24.4%)	164 (41.2%)
Attitude	24 (6.0%)	99 (24.9%)	275 (69.1%)
Perception	38 (9.5%)	120 (30.2%)	240 (60.3%)

**Table 6 healthcare-12-01596-t006:** Spearman’s correlation analysis between knowledge, attitude, and perception scores.

Variable	Rho */*p*-Value
Knowledge—Attitude	0.40 (0.001)
Knowledge—Perception	0.34 (0.001)
Attitude—Perception	0.69 (0.001)

* Significant value (two-tailed).

**Table 7 healthcare-12-01596-t007:** Associated factors of ASD-related knowledge among parents (*n* = 398).

Variables	Total	Knowledge			
		Low/Medium	High	Adjusted Odds Ratio (AOR) (95% CI)	*p*-Value
		*n* = 234	*n* = 164		
Age group					
18 to 30	207	119	88	Ref	
31 to 40	120	74	46	1.011 (0.59–1.74)	0.970
41 and above	71	41	30	0.593 (0.47–1.54)	0.593
Gender					
Male	201	120	77	Ref	
Female	197	114	87	0.841 (0.56–1.25)	0.395
Income					
Less than 5000 SAR	99	58	41	Ref	
5000 to 7000 SAR	105	57	48	1.122 (0.69–1.84)	0.645
More than 7000 SAR	194	119	75	1.336 (0.826–2.160)	0.237
Education status:					
Up to high school	153	83	70	Ref	
University/Graduate studies	245	151	94	1.355 (0.90–2.04)	0.146
Marital status					
Currently married	357	211	146	Ref	
Divorced	41	23	18	0.884 (0.46–1.68)	0.711
Occupation:					
Government	205	122	83	Ref	
Private	60	34	26	0.965 (0.62–1.50)	0.874
Unemployed	133	78	55	1.084 (0.59–2.01)	0.796
Children less than 10 years					
No	107	65	42	Ref	0.631
Yes	291	169	291	0.895 (0.57–1.41)	
Number of children					
≤3	358	213	145	Ref	
>3	40	21	19	0.752 (0.39–1.45)	0.395
Any child suffering from ASD in the family?					
No					
Yes	358	219	139	Ref	
	40	15	25	1.840 (1.13–2.63)	0.038
Youngest child age (in years)					
0 to 6	356	208	148	Ref	
More than 6	42	26	16	1.897 (0.49–7.27)	0.350

**Table 8 healthcare-12-01596-t008:** Associated factors of ASD-related attitude among parents (*n* = 398).

Variables	Total	Attitude			
		Low/Medium	High	Adjusted OR (95% CI)	*p*-Value
		*n* = 123	*n* = 275		
Age group					
18 to 30	207	69	138	Ref	
31 to 40	120	35	85	0.731 (0.40–1.33)	0.305
41 and above	71	19	52	0.887 (0.46–1.71)	0.721
Gender					
Male	201	74	127	Ref	
Female	197	49	148	1.760 (1.14–2.71)	0.010
Income					
Less than 5000 SAR	99	30	69	Ref	
5000 to 7000 SAR	105	33	72	0.977 (0.58–1.63)	0.287
More than 7000 SAR	194	60	134	1.46 (1.03–1.74)	0.007
Education status					
Up to high school	153	49	104	Ref	
University/Graduate studies	245	74	171	0.702 (0.59–1.42)	0.702
Marital status					
Currently married	357	109	248	Ref	
Divorced	41	14	27	1.180 (0.59–2.37)	0.871
Occupation					
Government	205	63	142	Ref	
Private	60	20	40	0.969 (0.60–1.56)	0.898
Unemployed	133	40	93	0.860 (0.45–1.65)	0.651
Children less than 10 years					
No	107	38	69	Ref	
Yes	291	85	206	0.749 (0.47–1.19)	0.749
Number of children					
≤3	358	106	252	Ref	
>3	40	17	23	1.757 (0.90–3.42)	0.633
Any child suffering from ASD in the family?					
No					
Yes	352	108	245	Ref	
	45	15	30	1.134 (0.58–2.19)	0.092
Youngest child age (in years)				Ref	
0 to 6	356	114	242	1.213 (0.34–4.22)	0.762
More than 6	42	9	33	2.971 (0.62–1.10)	0.170

**Table 9 healthcare-12-01596-t009:** Associated factors of ASD-related perceptions among parents (*n* = 398).

Variables	Total	Perception			
		Low/Medium	High	Adjusted OR (95% CI)	*p*-Value
		*n* = 158	*n* = 240		
Age group					
18 to 30	207	78	129	Ref	
31 to 40	120	49	71	1.282 (0.74–2.21)	0.374
41 and above	71	31	40	1.123 (0.62–2.03)	0.702
Gender					
Male	201	90	111	Ref	
Female	197	68	129	1.538 (1.03–2.31)	0.037
Income					
Less than 5000 SAR	99	38	61	Ref	
5000 to 7000 SAR	105	41	64	1.103 (0.67–1.81)	0.699
More than 7000 SAR	195	79	115	1.072 (0.66–1.74)	0.778
Education status					
Up to high school	153	52	101	Ref	
University/Graduate studies	245	106	139	1.481 (1.07–2.25)	0.046
Marital status					
Currently married	357	140	217	Ref	
Divorced	41	18	23	1.213 (0.63–2.32)	0.562
Occupation:					
Government	205	83	122	Ref	
Private	60	22	38	0.907 (0.62–1.52)	0.974
Unemployed	133	53	80	0.144 (0.62–2.14)	1.144
Children less than 10 years					
No	107	40	67	Ref	
Yes	291	118	173	1.142 (0.72–1.80)	0.567
Number of children					
Less than 3	358	138	220	Ref	
More than 3	40	20	20	1.594 (0.82–3.07)	1.594
Any child suffering from ASD in the family?					
No					
Yes	353	144	209	Ref	
	45	14	31	0.655 (0.33–1.27)	0.214
Youngest child age (in years)					
0 to 6	356	143	213	Ref	
More than 6	42	15	27	1.241 (0.37–4.14)	0.435

## Data Availability

The raw data supporting the conclusions of this article will be made available by the authors on the request.

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
