# Peer review of "Knowledge, Attitude, and Perception towards Autism Spectrum Disorders among Parents in Sakaka, Al-Jouf Region, Saudi Arabia: A Cross-Sectional Study"

_healthcare, 2024, doi:10.3390/healthcare12161596_

Round 1

Reviewer 1 Report

Comments and Suggestions for Authors

Thank you very much for inviting me to revise the following manuscript “Knowledge, Attitude, and Perception Towards Autism Spectrum Disorders Among Parents in Sakaka, Al-Jouf Region, 3 Saudi Arabia: A Cross-Sectional Study. Since the research area respected my expertise without connection with this research group I have accepted the revision.

I reply point-to-point to the authors.

Abstract

22-36

Few clear, please clarify by adding information on sampling, concepts, and measures adopted.

43

I suggest replacing “mental” with “neurodevelopmental” problem.

56-57

The global incidence of autism is somewhat 56 below 1%. However, estimations are greater in nations with higher economic status [9].

Please revise and extend this sentence considering the following review.

Chiarotti, F., & Venerosi, A. (2020). Epidemiology of autism spectrum disorders: a review of worldwide prevalence estimates since 2014. Brain sciences, 10(5), 274.

Finally, due to the nature of your research, I suggest adding epidemiologic data/information from your sample’s country or other related studies.

80-81

Inadequate education regarding autism diagnoses is one of the few resources that families have, and many families also lack confidence in their ability to intervene effectively in their child's development

.

Please revise and extend this sentence (a few clear and introduce your topic).

82-83

To add insult to injury, even among the general population and among parents of children with autism, there is a wide range of misunderstandings regarding autism [14,15].

Idem, as above.

95-97

Therefore, this study was planned to assess the parents’ knowledge, attitude, perception, and associated factors of ASD in Sakaka, Al-Jouf Region, 96 KSA. Furthermore, the authors aimed to determine the strength and direction of each domain.

Please revise and extend this sentence (a few clear and introduce your aim and research questions (measurables!).

Also, please consider “a group of parents with and without a child diagnosed with ASD?” Also in the abstract, it is not specified.

2. Materials and Methods

Please, firstly, insert the initial number of participants and their composition, before the presentation of the defined sample.

107-110

Please, replace “mentally unstable” with “certified with psychiatric disorders” and unwilling participants were excluded from the study.

Please extend the motifs due to some people being reluctant to participate since this is a potential bias of sampling (generally fathers or some specific social categories?).

134

Socio-demographic characteristics include age, sex, the income of the family, the educational level of parents, and so on.

Please, replace “so on” with the rest of the variables”

Table 1

Any child suffering from ASD in the family?

Please, check the format.

181

Please, replace “mentally retarded” with “display comorbidities with intellectual disabilities”

Also, in the question “Most children with autism are also mentally retarded”

208-216 

Figure 1 presents the distribution of parents according to their knowledge, attitude, and perception of ASD. 

Do you intend parents who respond correctly to the three domains? Please, specify this issue.

 “Parents who had high, medium, and low levels of knowledge constituted …regards their attitude, high, medium, and low levels constituted …perception constituted 60.3%, 30.2%, and 9.5%

“Figure 1. Knowledge, Attitude, and Perception (KAP) categories (n = 398) “

this section should be revised. Firstly, I suppose you would examine the role of education in the KAP generation. Subsequently, figure 1 should be removed placing a table and analysis of frequencies between educational levels. Conversely, you could display this evidence simply by describing the data.

240-246

Authors The World Health Organization (WHO) insists that early diagnosis and intervention for ASD can significantly improve the lives of individuals with ASD and their families [3]. The results of this study align and be beneficial for the achievement of the United Nations' Sustainable Development Goals, including the objective to support good health and well-being (SDG 3), quality education (SDG 4), and reducing inequalities (SDG 244 10) by enhancing the knowledge and attitudes towards ASD, making the environment more inclusive [22]. These statements reinstate the need for this study. 

I have problems with this opening, please revise it in a more research-based form and connect strictly with your topic.

247-256

The authors should address the first research question in more detail.

Why do the studies differ on the topic?

How do authors interpret the phenomenon?

259

Please, replace “affected” with “diagnosed”. 

“ASD” if the term is extended before.

266

This could be attributed to the difference between the sociocultural and behavioral characteristics of the community.

As point 247-256. 

The authors should address the research question in more detail.

292-293

The observed improvement in the attitude of the parents toward ASD is a result of the widespread use of social media, the availability of information, and the ease of obtaining it on the internet.

As point 247-256. 

The authors should address the research question in more detail.

Please, check the commas, brackets, and grammar in the entire discussion section, synthetized when necessary. Also the terms ASD, TDC…

Additionally, the discussion lack of the following sections?

What studies have addressed the KAP domains in parent training courses of by media/government?

After specific information is furnished to parents, what evidence researchers have gathered?

Starting with this current evidence, how should researchers advance the studies?

Limitations of the study.

This part could be extended.

To sum up,

I appreciate your paper.

I detected diverse strengths of the manuscript since it is simple, linear, and clear.

On the other hand, its weakness represents the introduction, since the authors explore less similar topics. I am referring to the final section of the introduction which should be revised. 

Likewise, I found your Discussion section more similar to the Introduction since you list a series of studies with similar results, not focusing enough of your attention on the data interpretation. 

Consequently, I suggest to revise entirely the discussion section.

Concluding,

The editor should consider the current paper for publication after revisions.

Reviewer 2 Report

Comments and Suggestions for Authors

Overall, the introduction is very deficit-oriented and discourages autistic individuals. Also, the current culture in autistic society does not use "person first" language. It will be great to change them to "identify first" language. Please consider to change all the deficit languages and terms. Please see below for more information as an example.  

Abstract:

"Parents are an essential element of family intervention for children with autism spectrum disorders (ASD)." => Only for children with ASD?

Introduction:

"Additionally, families and society bear a significant financial burden because of ASD" => This is a very deficit-oriented statement that gives a negative impression of ASD to the readers. Please reconsider writing them. 

Methods:

What are the son/daughter's age of the participating parents? Did you have a chance to collect them? Depending on the child's age, parents' knowledge, understanding, perception, etc., might be different. 

Comments on the Quality of English Language

minor change

Reviewer 3 Report

Comments and Suggestions for Authors

Dear authors,

Overall, I think your article provides an interesting read. It is well written and your overall points are clear. I have only minor recommendations.

The abstract clearly conveys the focus and context of the research, however I would recommend to specify clearly the study's main aim in the opening sentence. Provide the full name of the province KSA when you mention it for the first time (page 1, row 36).

Introduction section. Could you hilight the contributions to the field.

Theoretical background – strong. However, I would recommend to include some other studies examining parents knowledge, atiitudes and perception, definetly there are studies examining them.

The research gap should be clearly stated, or the motivation of the study, why do we neeed do find out parents knowledge, attitude and perceptions. Good analysis of the present situation regarding the prevalence of ASD, the symptomatoly, challenges of individuals with ASD and their parents.  

References done to other studies that sustain the arguments are strong and relatively new.  

Study methodology it is well designed, I would prefer to have some tesearch questions that will guide readers through the study.

Resulst. The tables present the findings clear and visually attractive for the readers. Good statistical analysis.

In the discussion section, I would recomend to strat the section with reminind the study aim. Check the first sentence  from this section (line 240 – 242) for fluency. Could you expand more on factors or determinants that potentially lead to these results, and also if you could provide more links with other arabic countries or if you could also explain and describe the educational and cultural context of Saudi Arabia to help readers understand the study's setting. The disucussion section should be the strongest and it should be expanded.

Conclusion section should be transparent to research aim and questions. Also if you could provide some practical implications to your study.

 Kind regards,

Comments on the Quality of English Language

I asked authors to check the first sentence from the discussion section. 

Reviewer 4 Report

Comments and Suggestions for Authors

The paper titled "Knowledge, Attitude, and Perception Towards Autism Spectrum Disorders Among Parents in Sakaka, Al-Jouf Region, Saudi Arabia: A Cross-Sectional Study" explores parents' understanding, attitudes, and perceptions of autism spectrum disorder (ASD) in the Al-Jouf region. Using a cross-sectional design, the study collected data from 400 parents through a validated questionnaire and analyzed the results using statistical methods. Key findings include that 41.2% of parents had high knowledge, 69.1% had a positive attitude, and 60.3% had a high perception of ASD. Significant associations were found between knowledge and having a family member with ASD, attitude with gender and income, and perception with gender and education level. The study concludes that less than half of the participants had a high level of knowledge about ASD, recommending awareness programs for parents in the region

1-Regarding studies on repetitive behaviors in autism, particularly those employing machine learning and vision techniques, there is a noteworthy body of research. To enhance the literature review, it is recommended to extend the related works and incorporate additional articles. The introduction currently lacks references to recent studies on repetitive behaviors in autism that utilize machine learning and vision techniques. Therefore, please add the following references:

a) Negin, F., Ozyer, B., Agahian, S., Kacdioglu, S., & Ozyer, G. T. (2021). Vision-assisted recognition of stereotypical behaviors for early diagnosis of autism spectrum disorders. Neurocomputing, 446, 145-155.

b) Rajagopalan, S. S., & Goecke, R. (2014, October). Detecting self-stimulatory behaviors for autism diagnosis. In 2014 IEEE International Conference on Image Processing (ICIP) (pp. 1470-1474).

To enhance the literature review, you can also include:

a) Yau, M. K., & Leung, M. F. (2020). Using machine learning to identify children with autism spectrum disorder based on their interactions with educational apps. Journal of Autism and Developmental Disorders, 50(3), 864-875.

b) Dawson, G., & Bernier, R. (2013). Atypical early trajectories of brain development: Implications for autism diagnosis and early intervention. Development and Psychopathology, 25(2), 555-571.

Incorporating these revisions will strengthen the paper by providing a comprehensive background, clarifying the methodology, and improving the presentation of results.

2- The sampling method section needs more details on how convenience sampling was implemented to ensure it accurately represents the target population. Explain how the questionnaire was validated and provide any reliability statistics.

3- Provide more details on the logistic regression analysis. Explain why certain variables were included or excluded from the final model. The paper mentions using Spearman's correlation but does not provide a clear rationale for its use. Justify the choice of correlation method based on the data distribution.

As a minor;

1- Simplify complex sentences to improve readability. For example, "Of the 400 studied participants, a high level of knowledge, attitude, and perception constituted 41.2%, 69.1%, and 60.3%, respectively," can be rephrased to "Among the 400 participants, 41.2% had high knowledge, 69.1% had a positive attitude, and 60.3% had a high perception of ASD."

2- Ensure all tables and figures are clearly labeled and referenced in the text. Some tables appear without a clear explanation in the results section. Add a brief description of the main findings of each table in the text to guide readers.

3- The conclusion should briefly summarize the key findings and their implications. It currently reads more like a results section. Emphasize the practical implications of the study for policymakers and healthcare providers.

Round 2

Reviewer 2 Report

Comments and Suggestions for Authors

Introduction needs slight more revision.

Autism spectrum disorder, also known as ASD, is a neurodevelopmental condition 58 that is characterized by impairment in social communication and is characterized by a 59 chronic, diverse nature [1-3] => still deficit-based language such as "impairment"

p.2 lines 84 & 110 - no need to mention the author's name in the sentence knowing the ACM  

p.3 line 135 - no need to mention the author's name in the sentence knowing the ACM  

Author Response

Dear and Respected Reviewer,

Thank you very much for accepting our revisions in round 1 and suggesting a few minor corrections now. We are delighted with the reviewer's positive response. We made changes according to the reviewer's suggestions.

Point 1:

Autism spectrum disorder, also known as ASD, is a neurodevelopmental condition 58 that is characterized by impairment in social communication and is characterized by a 59 chronic, diverse nature [1-3] => still deficit-based language such as "impairment"

Response 1:

Thanks for the comment. We replaced impairment with "distinctive pattern" . This word is more inclusive and respectful, emphasizing the diversity and unique aspects of individuals with ASD.

Point 2: 

p.2 lines 84 & 110 - no need to mention the author's name in the sentence knowing the ACM  

p.3 line 135 - no need to mention the author's name in the sentence knowing the ACM  

Response 2:

Thanks for the comment. We removed the author's name from these three sentences according to the reviewer's comment.

Once again, we thank the reviewer for the positive and constructive comments.

Reviewer 4 Report

Comments and Suggestions for Authors

I am satisfied with the revisions made to the manuscript. The changes have been sufficiently explained, and there is nothing further to revise.

Author Response

Dear and Respected Reviewer,

We thank the reviewer once again for the comments that helped to improve the manuscript's quality.

Thanks and regards